# Genomic and healthcare dynamics of nosocomial SARS-CoV-2 transmission

**Jamie M Ellingford[1,2]\*, Ryan George[3], John H McDermott[1,2], Shazaad Ahmad[4], Jonathan J Edgerley[1], David Gokhale[1], William G Newman[1,2], Stephen Ball[5,6], Nicholas Machin[4,7], Graeme CM Black[1,2]\***

[1]Manchester Centre for Genomic Medicine, St Mary's Hospital, Manchester University NHS Foundation Trust, Manchester, United Kingdom; [2]Division of Evolution and Genomic Sciences, School of Biological Sciences, University of Manchester, Manchester, United Kingdom; [3]Department of Infection Prevention & Control, Manchester University NHS Foundation Trust, Manchester, United Kingdom; [4]Department of Virology, Manchester Medical Microbiology Partnership, Manchester University NHS Foundation Trust, Manchester Academic Health Sciences Centre, Manchester, United Kingdom; [5]Division of Diabetes, Endocrinology & Gastroenterology, School of Medical Sciences, University of Manchester, Manchester, United Kingdom; [6]Department of Clinical Endocrinology, Manchester University NHS Foundation Trust, Manchester Academic Health Sciences Centre, Manchester, United Kingdom; [7]Manchester Medical Microbiology Partnership, Public Health England and Manchester University NHS Foundation Trust, Manchester, United Kingdom

**\*For correspondence:**
jamie.ellingford@manchester.ac.uk (JME);
graeme.black@manchester.ac.uk (GCMB)

**Competing interests:** The authors declare that no competing interests exist.

**Abstract** Understanding the effectiveness of infection control methods in reducing and preventing SARS-CoV-2 transmission in healthcare settings is of high importance. We sequenced SARS-CoV-2 genomes for patients and healthcare workers (HCWs) across multiple geographically distinct UK hospitals, obtaining 173 high-quality SARS-CoV-2 genomes. We integrated patient movement and staff location data into the analysis of viral genome data to understand spatial and temporal dynamics of SARS-CoV-2 transmission. We identified eight patient contact clusters (PCC) with significantly increased similarity in genomic variants compared to non-clustered samples. Incorporation of HCW location further increased the number of individuals within PCCs and identified additional links in SARS-CoV-2 transmission pathways. Patients within PCCs carried viruses more genetically identical to HCWs in the same ward location. SARS-CoV-2 genome sequencing integrated with patient and HCW movement data increases identification of outbreak clusters. This dynamic approach can support infection control management strategies within the healthcare setting.

## Introduction

The severe acute respiratory syndrome coronavirus-2 (SARS-CoV-2) pandemic continues to place a significant burden on healthcare services worldwide (*Miller et al., 2020*; *Propper et al., 2020*; *Maringe et al., 2020*). Reducing the spread and outbreak of SARS-CoV-2 infections is particularly important in hospitals and care homes where individuals at high risk of developing severe responses to infection are vulnerable to transmission due to close and regular contact between patients and healthcare workers (HCWs) (*Clark et al., 2020*; *Nguyen et al., 2020*; *Rivett et al., 2020*). Whilst the recent development of promising SARS-CoV-2 vaccines may reduce risk to individuals in hospitals who receive the vaccine (*Walsh et al., 2020*; *Anderson et al., 2020*; *Keech et al., 2020*;

*Lodge, 2020*), risk of infection will not be completely mitigated. Standard methods of infection control will continue to be required to ensure patient and HCW safety. Regular and iterative testing for SARS-CoV-2, in both patients and HCWs, underpins approaches that quantify and control nosocomial transmission (*Black et al., 2020*) but will not provide insights into how the virus may have spread throughout an institution, alone. Since the degree to which different groups (patients, HCW) propagate SARS-CoV-2 transmission remains uncertain, the utility of screening approaches to prospectively prevent nosocomial spread is difficult to evaluate. It is well established that HCWs are an important component in pathogen outbreak investigations (*Peacock et al., 2018*; *Wenger et al., 1995*; *de Swart et al., 2000*). Here, we have integrated viral genome sequencing with patient admissions records and staff workplace information to investigate SARS-CoV-2 nosocomial outbreaks. We demonstrate that such an approach can be used to identify highly likely nosocomial transmission events of SARS-CoV-2 between HCWs and the patients in their care.

## Results

### Sample demographics

We generated high-quality sequencing datasets for 173 samples that had been collected from inpatient wards, accident and emergency departments (A and E) and from HCWs across geographically distinct hospitals. All samples were collected between calendar week 11 (commencing 8 March 2020) and calendar week 23 (ending 6 June 2020). Of the 173 high-quality sequenced samples, 39 (23%) were HCW samples. The remaining 134 samples were collected from patients admitted to a total of 31 wards and units situated across the five hospitals. Forty-four (25%) of the 173 high-quality samples were from three hospital locations, which had seen sudden rises in SARS-CoV-2-positive cases; an additional 35 samples from these locations failed to meet our quality criteria for sequencing quality (*Figure 1—figure supplement 1*). Forty-seven (35%) of 134 patient samples were from A and E departments. The median age of the 134 patients with sequenced samples was 81 years (mean = 75 years; range = 6 weeks–100 years).

### Sequencing metrics

Comparison of SARS-CoV-2 amplicon yield versus target coverage found that samples consistently met $10\times$ minimum coverage thresholds when a total yield of >400 ng was obtained (median = 846; range = 86–3667; *Figure 1—figure supplement 1*). Where RT-qPCR cycle threshold (Ct) values were provided, we found samples with Ct values of 31 or more resulted in lower amplicon yield and frequently failed to meet sequencing coverage thresholds (*Figure 1—figure supplement 1*). In the 173 high-quality samples, we identified 268 genomic variants in comparison to MN908947.3, of which 86 were recurrent variants across samples (range = 2–126 samples) and 182 were unique to single samples.

### Global lineage assignment

We utilised Pangolin for placement of the 173 high-quality viral genome sequences within the global SARS-CoV-2 phylogenetic tree. Eight-seven percent (151/173) of sequenced genomes were confidently assigned to an existing lineage (SH-alrt > 80%, UFbootstrap > 90%). We identified 11 distinct lineages in our cohort, with a bias towards viral genomes assigned to lineage B.1.1 (71%, 122/173; *Figure 1—figure supplement 2*). There were no samples assigned to lineage A. Incorporating the calendar week of sample collection into this analysis suggested a constant relative frequency of viral lineage B.1.1 over time (*Figure 1—figure supplement 2*). Viral lineage B.1.1 was present in samples collected from 29 different wards, including A and E, and in HCWs, and was present in our cohort between calendar weeks 12–23 (*Figure 1—figure supplement 2*).

### Local phylogenetic networks implicate nosocomial transmission within hospital wards

To understand how the viral genomes collected from different areas across hospital sites related to one another we created a local phylogenetic tree rooted to a SARS-CoV-2 genome originally sequenced in Wuhan, China (MN908947.3) (*Minh et al., 2020*; *Wu et al., 2020*). We overlaid locational origins of the samples (i.e. ward and units from which samples were collected) onto the

phylogenetic tree (*Figure 1—figure supplement 3*). We observed clusters within the phylogenetic tree that were formed predominantly from viral genome sequences taken in individual wards (*Figure 1—figure supplement 3*). For example, 19/31 of the samples in one of these identified clusters were collected from a single hospital ward (*H2_W7*, *Figure 1—figure supplement 3*). The phylogenetic clusters were supported by maximum-likelihood and consensus (10,000 ultrafast bootstraps) approaches for tree creation.

## Incorporating staff and patient movement enhances identification of nosocomial transmissions

### Patient contact clusters

Utilising patient admissions data over the period of the pandemic, we first created a network of potential direct or indirect patient–patient contacts, inferred through the presence of two individuals on the same ward on the same calendar day. We adopted national guidelines for definition of nosocomial infection to identify contacts between individuals relative to the date of sample collection for a positive SARS-CoV-2 test. Using a contact window of 3–7 days prior to a positive SARS-CoV-2 test (termed herein as *likely* period of infection), we developed a network of patient–patient contacts (*Figure 1*). We identified eight significant clusters of individuals (patient contact clusters [PCCs]), defined by multiple potential contacts between two or more individuals within the likely period of infection for each individual (A–H, *Figure 1*). We assessed the pairwise similarity of viral genomes in identified clusters, demonstrating significantly higher viral genetic similarity within clusters compared to non-clustered samples (*Figure 2*, p<0.001). This trend was further supported by overlaying the PCCs onto the local phylogenetic tree (*Figure 1—figure supplement 4*). We identified areas of the phylogeny where there was a 20-fold increase, than expected by chance, in potential contacts between a patient and six or more patients with the most closest genetically related viral genome samples. Consecutive windows of increased (20-fold) patient–patient contacts were merged to identify seven distinct clusters of individuals defined by high level of genetic relatedness of the viral genome sequence and a high degree of potential patient–patient contacts during the likely period of infection (*Figure 1—figure supplement 4*).

### Contact clusters including patients and HCWs

Next, we created networks of potential interactions between HCWs and patients (*Figure 1*). This was inferred through the presence of patients in the direct workplace of staff members for at least one calendar day in the 7 days prior to a patient testing positive for SARS-CoV-2. We identified 49 potential contacts between HCWs and patients, including 10 HCWs and 18 patients. Incorporating this information onto the local phylogeny expanded the density of contacts within hotspots (*Figure 1—figure supplement 4*) and altered the structure of the PCCs within the likely period of infection (*Figure 1*). Overall, we identified significantly increased genetic similarity in viral samples between patients and HCWs in the same ward locations in comparison to patients and HCWs from different wards (*Figure 2*, p<0.001). Moreover, we observed greater genetic similarity within each of the PCCs with HCW interactions incorporated (*Figure 2*, p<0.001). For example, HCW_A illustrates that the addition of HCWs created a previously hidden link between PCC_A and PCC_C (*Figure 1*); these newly identified connections increased the number of individuals within viral clusters (VCs), defined as clusters of identical viral samples or viral samples that differed by just a single genomic variant (*Figure 1*). Within the newly defined contact clusters including HCWs and patients, we identified six genetically identical VCs including 18 individuals, and 24 individuals with viral sequences differing by a single genomic variant. The number of individuals in VCs was expanded by including patient–HCW contact networks, over and above those identified using patients alone (*Figure 1*).

## Temporal patterns within identified patient and HCW contact clusters

In order to establish the most likely SARS-CoV-2 transmission pathways, we examined temporal patterns within proposed nosocomial outbreaks. We created a median joining network for each of the patient and HCW contact networks and incorporated time of sample collection. As multiple entry points into the outbreaks may complicate inference, we cleaned the data to create median joining networks for the 10 most genetically related viral samples within each cluster (inferred from position in local phylogeny, *Figure 1—figure supplement 3*). This identified trends in the datasets that

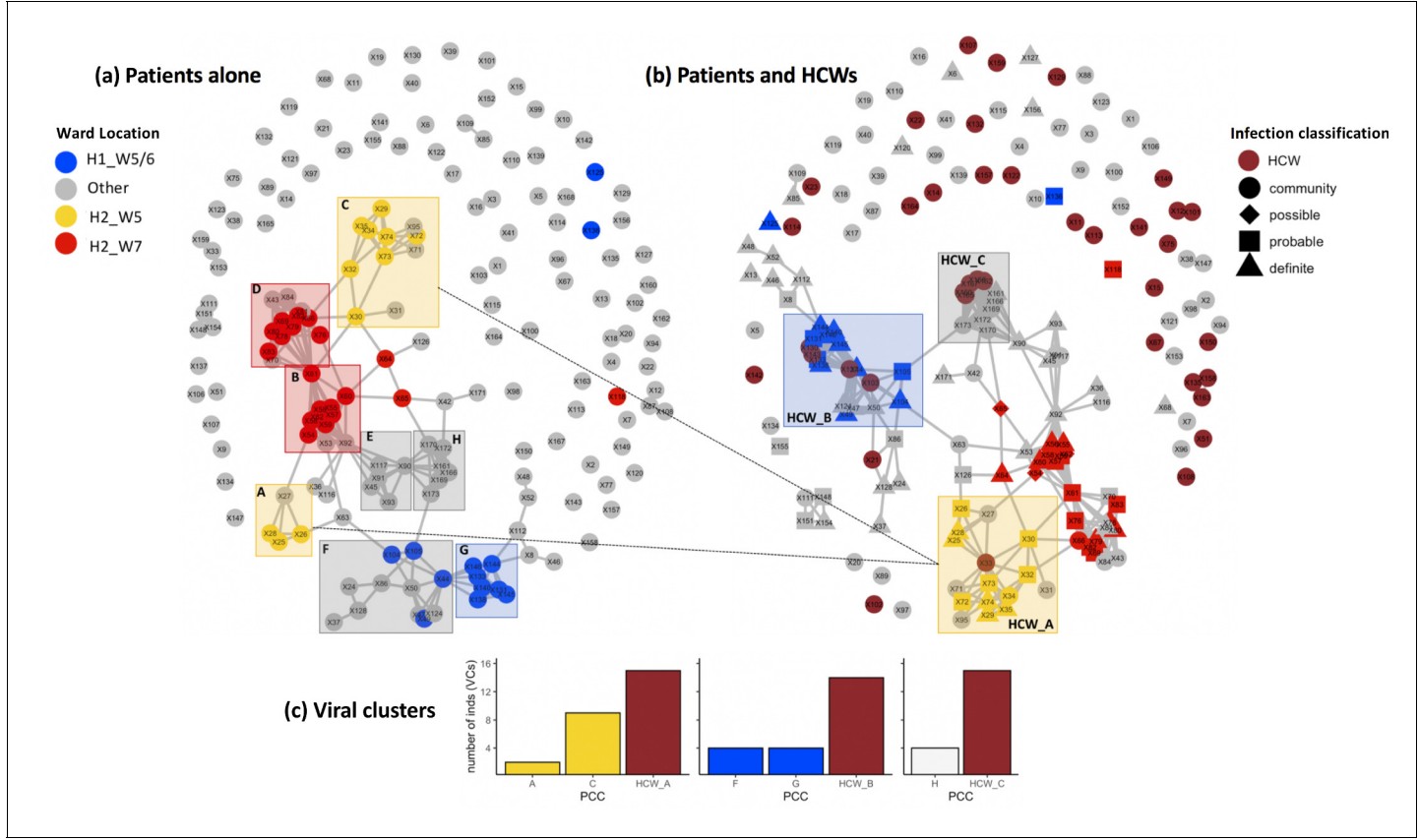

**Figure 1.** Incorporating healthcare worker (HCW) and patient admissions data into the analysis of viral genetic relatedness improves certainty of nosocomial outbreaks. (a) Network of direct and indirect potential patient–patient contacts within the window of likely infection (3–7 days prior to positive SARS-CoV-2 test) defines eight significant patient contact clusters (PCCs, *overlaid boxes*); (b) network including HCW interactions one week prior to positive SARS-CoV-2 test and patient infection classification. Nodes represent individual patients or HCWs, with ordinal numbers representing their position in the constructed local phylogenetic tree. Edges indicate presence on the same hospital ward on the same calendar day. Inclusion of HCWs brings together originally disparate PCCs (b) and (c) increases the number of individuals within viral clusters (VCs) – defined as clusters of identical viral samples or derived viral samples which differ by a single genomic variant. We identified 44 individuals within VCs in the newly defined HCW contact clusters (HCW_A, HCW_B, HCW_C), 21 of whom were not identified within VCs using PCCs alone. The shape of symbols within the enlarged boxes displays the classification of SARS-CoV-2 infection in patients: *community*, community-acquired infection (positive test within 2 days of hospital admission); *possible*, possible hospital-acquired infection (positive test 3–7 days after hospital admission); *probable*, probable hospital-acquired infection (positive test 8–14 days after hospital admission); *definite*, definite hospital-acquired infection (positive test >14 days after hospital admission). The presence of several patients with definite and probable hospital-acquired infections within the PCC and HCW interaction clusters further reinforces the risk of SARS-CoV-2 transmission events between patients and HCWs on the same hospital wards.

The online version of this article includes the following figure supplement(s) for figure 1:

**Figure supplement 1.** *Top*. The yield of DNA (ng) from tiled PCR enrichment is a good indicator of the amount of the SARS-CoV-2 genome covered by at least 10 sequencing reads.

**Figure supplement 2.** Global lineage assignment of 173 SARS-CoV-2 genomes sequenced across Manchester University Hospital Foundation Trust.

**Figure supplement 3.** Local phylogenetic tree created for 173 SARS-CoV-2 genomes sequenced across Manchester University Hospital Foundation Trust.

**Figure supplement 4.** Incorporating patient admission data into the analysis of phylogenetic relationship identifies hotspots of contacts between patients and healthcare workers (HCWs).

further reinforced the likelihood of nosocomial infection (*Figure 3*). We observed that all identified contact clusters could be rooted back to potential 'founder' viral samples that occurred early during the suspected outbreak. For example, in PCC_B and PCC_D (*Figure 3*), which occurred on the same hospital ward at different time points, the original 'founder' viral samples contained the novel (at time of sample collection) genomic variant MN908947.3–3228-T-G. Eighteen (95%) of 19 viral samples subsequently collected from this hospital ward also contained the same novel variant, at least 16 of these cases were probable or definite hospital-acquired infections. Reducing the quality

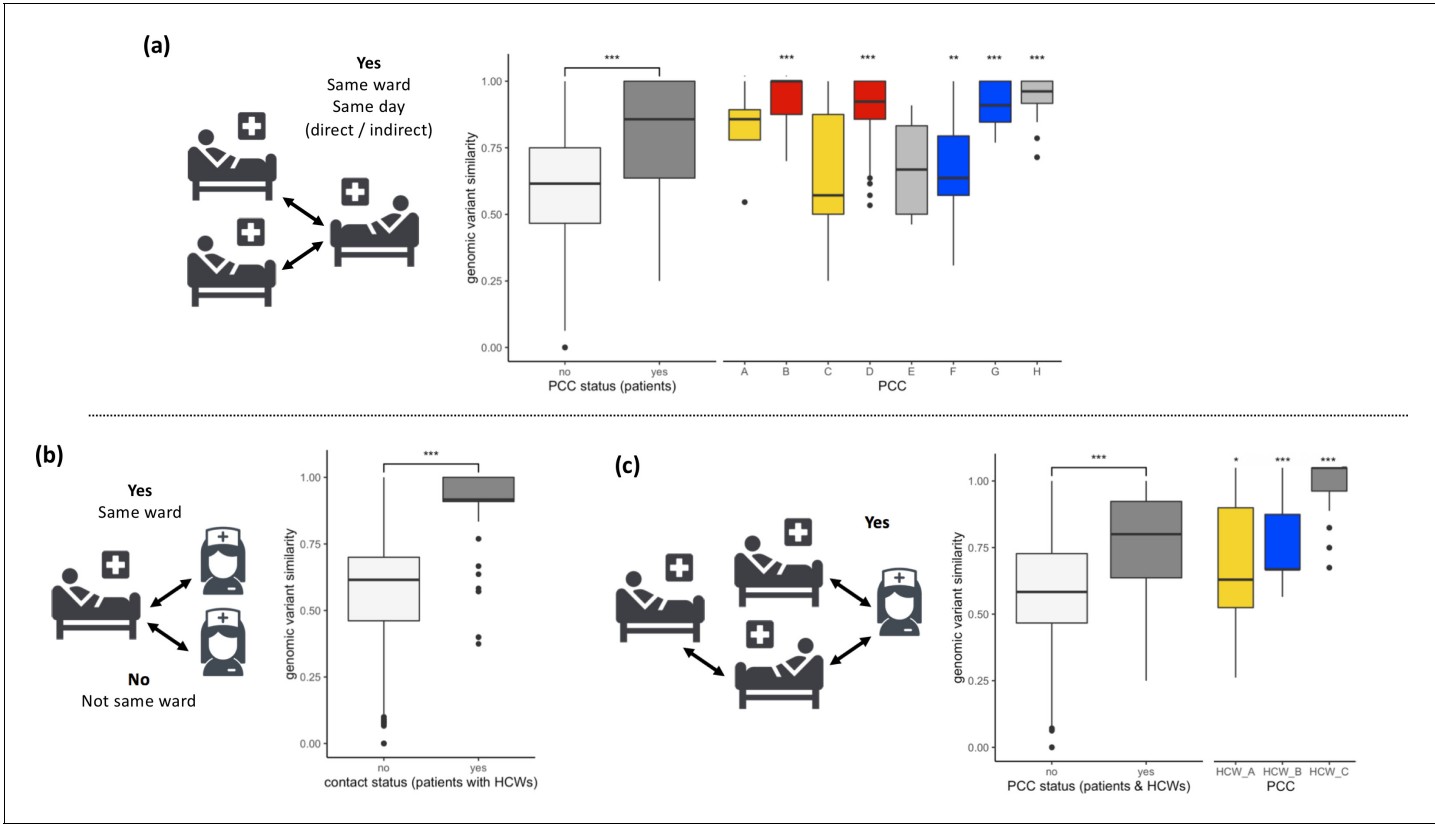

**Figure 2.** SARS-CoV-2 viral genomes are more similar in groups of patients and healthcare workers (HCWs) who have been in contact prior to a positive SARS-CoV-2 test. (**a**) Pairwise genomic variant similarity comparisons of SARS-CoV-2 genomes by the status within patient contact clusters (PCCs) demonstrates increased genetic similarity when patients have been in direct or indirect contact with one another 3–7 days prior to positive SARS-CoV-2 test. Pairwise comparisons within PCCs (n = 544) were tested against all pairwise comparisons that were not defined within the PCCs (n = 29,212). (**b**) Pairwise genomic variant similarity comparisons of SARS-CoV-2 genomes by patient–HCW interactions in the week prior to positive SARS-CoV-2 test (n, yes = 98, no = 11,836). (**c**) Pairwise genomic variant similarity comparisons of SARS-CoV-2 genomes by presence within PCCs including interactions with HCWs. Pairwise comparisons within HCW clusters (n = 846) were tested against pairwise comparisons that were not defined within the PCCs including HCW interactions (n = 28,168). Colours of boxplots reflect the PCCs identified in *Figure 1*, and asterisks indicate significance level determined through a two-sided Wilcoxon rank-sum test (*<0.05; **<0.01; ***<0.001 after Bonferroni correction for multiple testing).

threshold for sequencing datasets (≥50% coverage at ≥10× coverage) identified the MN908947.3–3228-T-G variant in an additional five samples collected from this hospital location (83%, n = 6 additional samples included with modified criteria). These data highlight that samples excluded from our analyses due to sequencing quality criteria may be missing links within SARS-CoV-2 transmission pathways.

We observed further trends indicative of nosocomial outbreaks, including the number of genomic variants identified against the MN908947.3 reference genome increasing over time within each of the contact networks, and consistently observed that samples collected early during the suspected outbreaks had a greater number of derived or identical samples within the outbreak than those collected at a later day (*Figure 3*). Both of these trends would be expected if single ancestral SARS-CoV-2 sequences were original founders of an outbreak.

## Discussion

In this study, we obtained high-quality SARS-CoV-2 genomic sequences for 173 individuals across five hospitals in the North West of the UK, including both patients and HCWs. We incorporated potential contacts between the two groups into a phylogenetic analysis and comparison of pairwise genetic similarity. Overall, this demonstrated that inclusion of contact data increased confidence in the characterisation of nosocomial outbreaks.

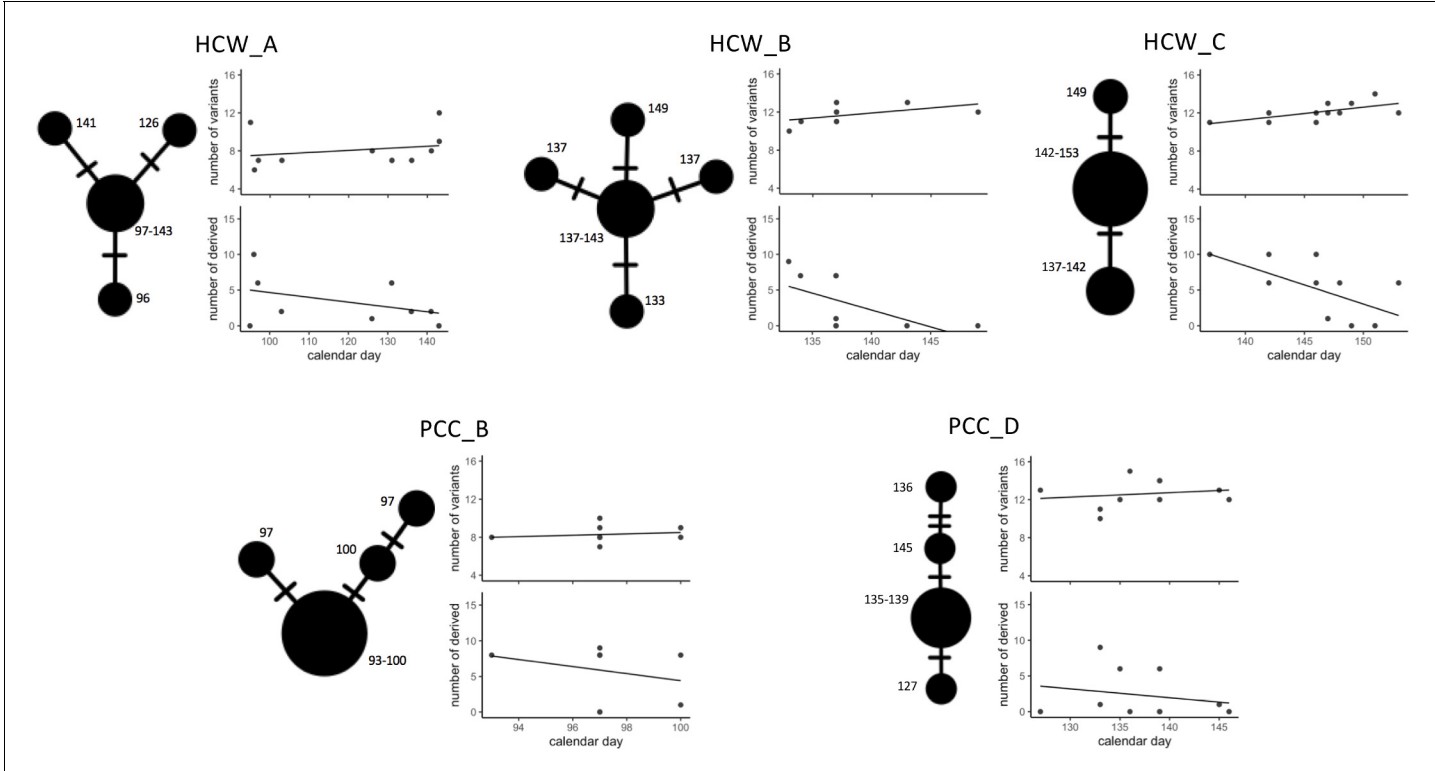

**Figure 3.** Temporal patterns in SARS-CoV-2 genomic similarity identify potential viral transmission pathways within patient contact clusters (PCCs) including healthcare worker (HCW) interactions. For each of the highlighted contact clusters, a median joining network is presented with *size* of nodes representing number of samples and *numbers* indicating day of nasal or throat swab collection. The presented network suggests a possible path of viral transmission within each contact cluster, *hatches* represent single genomic variants that differ between viral clusters. The *top* scatterplot shows that the number of genomic variants identified against the MN908947.3 reference genome increases over time. The *bottom* scatterplot shows the number of other samples within the contact cluster that are identical or expected to be derived from samples collected at specific calendar days – these are defined as other samples that are identical but with the presence of additional genomic variants. The observed trends show that samples collected early during the suspected outbreaks have a greater number of derived or identical samples than those collected at a later day. These data support that the samples collected early during the highlighted contact clusters are early founder events during a nosocomial outbreak.

This work was undertaken across multiple, geographically separate hospitals in the UK with responsibility for the care of large numbers of SARS-CoV-2-positive patients. At the time of analysis, there was no routine SARS-CoV-2 screening of asymptomatic HCWs. Similar to other UK hospitals, the different locations within the hospitals were assigned as either green (SARS-CoV-2-negative) or red (SARS-CoV-2-positive) zones. This strategy, in combination with additional infection control measures such as staff bubbles, is widespread as a method to reduce nosocomial infection. However, as patients tested positive after spending prolonged periods of time in green areas, it became apparent that there were unrecognised transmission events between the two areas or from the community into green zones.

Viral genome sequencing offers a realistic possibility to track and identify root causes of nosocomial transmissions (*Lucey et al., 2020*; *Meredith et al., 2020*). Here, we applied genome sequencing to throat and nasal swabs obtained from HCWs and patients. We identified 268 unique genomic variants in the 173 high-quality samples and placed our samples within recognised global lineages (*Figure 1—figure supplement 2*). The predominance of a single SARS-CoV-2 lineage meant that precise differentiation of viral samples from individual hospital wards was not possible at this level (*Figure 1—figure supplement 2*). We therefore created a local phylogeny for our sequenced genomes rooted to MN908947.3 (*Figure 1—figure supplement 3*) and calculated pairwise similarity in the genomic variants identified in each of the SARS-CoV-2 genomes. It has been previously noted that the low genetic diversity of SARS-CoV-2 causes complexity in the identification of nosocomial outbreaks, as samples may be genetically identical by chance rather than through transmission

between individuals (*Meredith et al., 2020*; *Gudbjartsson et al., 2020*). Our data reemphasise this low genetic diversity of SARS-CoV-2, with a median number of 11 (range = 2–16) variants identified per sample, and an average pairwise similarity of 61.5% (*Figure 2*). Integrating our analyses with patient admissions record (available for 104/134 patients for at least one day prior to SARS-CoV-2-positive test) and location of staff workplaces (available for 31/39 HCWs) identified clusters of individuals who had interacted during their most likely periods of infection. These analyses confirmed that individuals who had been in contact during this period were more likely to have genetically more similar viral samples than individuals who had not been in contact (*Figure 2*).

While we cannot exclude that viral samples are genetically identical or similar by chance due to the low genetic diversity of SARS-CoV-2, the spatial and temporal patterns of viral genetic relatedness that we observe provide strong evidence for nosocomial transmission amongst both patients and HCWs. These trends are observed in at least five distinct clusters, across three geographically distinct hospitals. These are further reinforced by the presence of novel genomic variants (at the time of analysis) transmitting through identified clusters (*Figure 1—figure supplement 3*). We suggest these data support the widespread adoption of iterative screening strategies for HCWs who may be pre-symptomatic or asymptomatic shedders of SARS-CoV-2 (*Black et al., 2020*; *Arons et al., 2020*; *Buitrago-Garcia et al., 2020*). Pre-symptomatic or asymptomatic individuals have been demonstrated to be important contributors to SARS-CoV-2 outbreaks (*Rivett et al., 2020*; *Kasper et al., 2020*; *Letizia et al., 2020*).

Others have shown that comprehensive characterisation of outbreak clusters can be hindered by the existence of hidden links between individuals, even where all individuals within clusters are known and completely isolated from external contacts (*Sekizuka et al., 2020*). In our data, we observed the presence of cohort-specific genomic variants shared between individuals but without a known connection between the sampled individuals. Here, it is likely that missing individuals or connections between surveyed individuals are adding complexity to our analyses. While the use of digital contact tracing is difficult in hospital environments, track and tracing smartphone software could be useful to extend the characterisation of contacts between individuals and to understand the accuracy of the assumptions enforced in this study (*Firth et al., 2020*; *Ferretti et al., 2020*). Here, we infer contacts between individuals through their presence on the same hospital ward on the same calendar day. This approach identifies individuals within our cohort that are likely to have been in face-to-face contact. We note that this approach has imperfect assumptions but is likely to dilute rather than inflate the statistical significance of the investigations reported in this study (*Figures 1–3*).

Future work may enable additional co-factors to be considered in models for network creation such as infection control measures in place on hospital wards (e.g. personal protective equipment utilised), symptomatic status of individuals, and the length, type, and the proximity of physical contacts between individuals. These approaches are supported by recent data demonstrating that over half of SARS-CoV-2 transmissions occur when individuals are pre-symptomatic and that transmission likelihood increases with the duration and proximity of contact (*Sun et al., 2021*). Collecting data to incorporate these factors into network models in the healthcare setting may enable the generation of more precise binary contact clusters according to specified parameters or the development of weighted networks biased by the relative importance placed on co-factors (*Firth et al., 2020*). Understanding the concordance of empirical datasets of SARS-CoV-2 transmission, as reported here, and computational models of transmission is an important avenue for future work to identify the most influential factors to decrease the likelihood of SARS-CoV-2 transmission in both healthcare and community settings.

Our data demonstrate that SARS-CoV-2 genome sequencing alongside patient admission and staff workplace information can identify transmission events within the healthcare setting. Looking forward, we expect that the adoption of genomic approaches in real time, for example within 48 hr, alongside consideration of patient movement datasets will enable rapid identification of linked hospital-acquired SARS-CoV-2 infections. Such approaches could optimise infection control management strategies, lead to targeted interventions, reduce nosocomial transmission, and ultimately prevent avoidable harm to vulnerable individuals who acquire COVID-19 whilst in the healthcare setting.

# Materials and methods

## Sample selection

Throat and nasal swab samples were collected from patients and healthcare professionals based at MFT hospital sites. Diagnostic SARS-CoV-2 RT-qPCR assays were performed by the Clinical Virology Department of the Manchester Medical Microbiology Partnership (MMMP; Manchester, UK). RT-qPCR-positive samples were selected for SARS-CoV-2 whole-genome analyses at the Manchester Centre for Genomic Medicine (MCGM; Manchester, UK). We attempted to sequence all available SARS-CoV-2-positive samples from hospital wards highlighted by infection control surveillance officers as potential outbreaks within our sample collection period due to sudden rises in positive cases. Demographic, hospital location, and laboratory data were included with each referral. All ward names have been anonymised for publication.

## Sample and NGS library preparation

Nucleic acid re-extraction was performed using the chemagic Viral DNA/RNA 300 Kit on the chemagic 360 instrument (PerkinElmer Inc, Waltham, MA). All extracted RNA samples underwent cDNA synthesis using either LunaScript RT SuperMix kit (New England Biolabs, Ipswich, MA) or SuperScriptIV (Thermo Fisher Scientific, Waltham, MA), in accordance with manufacturers protocols. SARS-CoV-2 whole-genome libraries were prepared using SureSelectXT Low Input kit CoVHuman6X enrichment capture-based method (Agilent Technologies, Santa Clara, CA) or the ARTIC tiled amplicon multiplex PCR protocol (version three primer set) with NEBNext Ultra II DNA Library Prep Kit (New England Biolabs). PCR and library preparation quality validations were obtained using TapeStation D1000 and HSD1000 (Agilent Technologies). Final libraries were sequenced using MinION flow cells version 9.4.1 (Oxford Nanopore Technologies, Oxford, UK) or MiSeq (Illumina, San Diego, CA) using reagent kits for 600 cycles (for tiled PCR SARS-CoV-2 amplification) or 300 cycles (for Agilent SureSelectXT enrichments) for paired end sequencing.

## Bioinformatics and analysis

Sequencing reads were deduplicated on instrument for Illumina MiSeq datasets or using Guppy for Oxford Nanopore datasets. Reads were aligned to the SARS-CoV-2 reference genome (MN908947.3) using BWA-MEM (*Li, 2013*) for Illumina MiSeq datasets and using Minimap2 (*Li, 2018*) for Oxford Nanopore MinIon datasets. Reads were filtered and variants identified using iVar v1.2.2 (*Grubaugh et al., 2019*). Samples with ≥75% of the MN908947.3 reference genome covered by ≥10 high-quality reads with at least 50 aligning nucleotides were included for downstream analysis. Variants with an allele fraction of at least 0.6 in high-quality mapped reads were identified in comparison to MN908947.3, and a consensus FASTA built using iVar. Multi-way alignments were performed using MAFFT v7.407 (*Katoh et al., 2002*), and maximum-likelihood trees rooted to MN908947.3 using 1000 bootstraps were generated with IQ-TREE v1.6.12 (*Minh et al., 2020*). Trees were visualised in Geneious Prime software v2020.1.2 (https://www.geneious.com). Pangolin v2.0 (*Rambaut et al., 2020*) was utilised for positioning of sequences within the global phylogenetic tree (lineages v2020-05-19). Median joining networks were created in Pop-ART (http://popart.otago.ac.nz/). Pairwise similarity analyses were performed using a bespoke script and calculated the number of exact matches in genomic variants between samples after adjusting for regions masked by low coverage. All high-quality genome sequences were shared with COG-UK (*COVID-19 Genomics UK (COG-UK), 2020*).

## Patient admissions and movement

We collected hospital admission data for all patients with high-quality sequenced genomes. For each patient, we identified other individuals within the cohort who were present on the same hospital wards on the same calendar day, leading to potential indirect or direct contacts between patients. This method for defining contacts assumes that close face-to-face contact is the most likely method for SARS-CoV-2 transmission between individuals and aims to identify individuals within our cohort who are most likely to have had such interactions. This assumption is supported through recent meta-analyses concluding that physical distancing of less than 1 m increases likelihood of SARS-CoV-2 transmission between individuals (*Chu et al., 2020*). We assessed all potential contacts

for each patient in relation to the calendar day that the positive SARS-CoV-2 nasal or throat sample was collected from the patient. The windows of contacts are defined in accordance with national guidelines for SARS-CoV-2 nosocomial outbreak: community-acquired infection (positive test within 2 days of hospital admission); possible hospital-acquired infection (positive test 3–7 days after hospital admission); probable hospital-acquired infection (positive test 8–14 days after hospital admission); and definite hospital-acquired infection (positive test >14 days after hospital admission). For patient–patient contacts, we identified any potential contacts within the possible hospital-acquired infection period (3–7 days) and developed a binary matrix. We made additional assumptions for including HCWs in the contact networks, specifically, we assumed that HCWs had direct or indirect contacts with all patients who had been present in their workplace (hospital ward) and assumed interaction between HCWs and patients were constant up to the day of positive SARS-CoV-2 tests for patients – this extended the contact window incorporated into the binary matrix to include any interactions between patients and HCWs in the 1–7 days prior to a patient testing positive for SARS-CoV-2. Networks of potential patient–patient and HCW contacts were constructed using the ggnet and ggplot packages in R. PCCs were identified as discrete areas of the contact networks that included three or more interconnected nodes where the density of edges outnumbers the number of nodes. Outliers connected to the PCCs were also included if they were not connected to any other PCC. Nodes acting as connecting hubs between distinct clusters were assigned to one of the otherwise mutually exclusive PCCs. Pairwise viral similarity was compared between clusters of the network using a two-sided Wilcoxon rank-sum test.

## Additional information

### Funding

| Funder | Grant reference number | Author |
| --- | --- | --- |
| Health Education England | | Jamie M Ellingford |
| Manchester NIHR Biomedical Research Centre | IS-BRC-1215-20007 | William G Newman |

The funders had no role in study design, data collection and interpretation, or the decision to submit the work for publication.

### Author contributions

Jamie M Ellingford, Conceptualization, Data curation, Formal analysis, Investigation, Visualization, Methodology, Writing - original draft, Writing - review and editing; Ryan George, John H McDermott, Jonathan J Edgerley, Data curation, Formal analysis, Investigation, Writing - review and editing; Shazaad Ahmad, Investigation, Writing - review and editing; David Gokhale, Data curation, Investigation, Methodology, Writing - review and editing; William G Newman, Stephen Ball, Nicholas Machin, Supervision, Investigation, Writing - review and editing; Graeme CM Black, Conceptualization, Supervision, Investigation, Writing - review and editing

### Author ORCIDs

Jamie M Ellingford (iD) https://orcid.org/0000-0003-1137-9768

### Ethics

Human subjects: The study was conducted to investigate hospital outbreak investigation/surveillance; individual patient consent or ethical approvals were not required. The study protocol was approved by the Manchester Biomedical Research Centre COVID-19 rapid response group and the Manchester University NHS Foundation Trust Executive Committee. All samples and data collected were part of routine care or hospital operational policy. No patient-identifiable/individual identifiable data are presented.

### Decision letter and Author response

Decision letter https://doi.org/10.7554/eLife.65453.sa1

Author response https://doi.org/10.7554/eLife.65453.sa2

## Additional files

### Supplementary files
• Transparent reporting form

### Data availability

All genome sequencing datasets have been shared with COG-UK21. Instructions for data access are provided at https://www.cogconsortium.uk/tools-analysis/public-data-analysis-2/.

The following previously published dataset was used:

| Author(s) | Year | Dataset title | Dataset URL | Database and Identifier |
|---|---|---|---|---|
| Wu F, Zhao S, Yu B, Chen YM, Wang W, Song ZG, Hu Y, Tao ZW, Tian JH, Pei YY, Yuan ML, Zhang YL, Dai FH, Liu Y, Wang QM, Zheng JJ, Xu L, Holmes EC, Zhang YZ | 2020 | Severe acute respiratory syndrome coronavirus 2 isolate Wuhan-Hu-1, complete genome | https://www.ncbi.nlm.nih.gov/nuccore/MN908947 | NCBI GenBank, MN908947.3 |

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
