## [Decision Letter]

**Acceptance summary:**

This work describes how contextual information can be used together with genomic sequencing data to better understand nosocomial SARS-CoV-2 transmission in healthcare settings. This work is of importance to health professionals in charge of tracking nosocomial infections and monitoring infection control strategies.

**Decision letter after peer review:**

Thank you for submitting your article "Genomic sequencing and healthcare dynamics track nosocomial SARS-CoV-2 transmission" for consideration by *eLife*. Your article has been reviewed by two peer reviewers, including Niel Hens as the Reviewing Editor and Reviewer #1, and the evaluation has been overseen by a Senior Editor. The following individual involved in review of your submission has agreed to reveal their identity: Josh Firth (Reviewer #2).

Essential Revisions:

1) In their study the authors analyse data from 173 high-quality SARS-CoV-2 genomes. They identified eight patient contact clusters based on these 173 SARS-CoV-2 genomes and compare their results with an analysis taking into account patient and HCW movement data which impacts the identification of individuals belonging to the contact clusters. It is based on these differences that the authors conclude that the patient and HCW movement data improve the understanding of the role of patient-HCW interactions. I believe the authors should provide a more thorough assessment of this claim. I would like to note that their endeavour is, as such, not that novel and that researchers have looked into identifying transmission paths based on genomic surveillance with or without contextual information (see references included in the manuscript) also highlighting the importance of issues related to gathering the data needed for doing so.

2) It is not clear from the manuscript whether truly all patients had samples that were subject to genomic sequencing. How (un)likely is the exclusion of data on patients and HCW that were not sequenced and what impact might be expected from their omission?

3) The authors indicate that their work can lead to better tracking and infection control which I believe to be true but they do not provide any specific guidance on how this can be done. This final paragraph of the Discussion may be a useful place to flesh out some ideas about how this empirical data on COVID19 spread within a real-world setting may also be useful of parameterising models aimed at predicting COVID19 spread and control efforts.

4) Subsection “Patient Admissions and Movement”. Some additional mention of the assumptions made here (in terms of inferring contacts in this way) would be useful. Specifically, as other studies have shown that this disease is most likely to spread by short-range face-to-face contacts, it would be interesting to hear how these inferred contacts here might align to types of contact we would expect to be relevant to transmission.

5) The text reads "Pairwise viral similarity was compared between clusters of the network" but there is no mention of how “clusters” within the network were defined/found. Subsection “Patient contact clusters”, again, difficult to interpret how these “clusters” were quantified from the network data. Subsection “Contact clusters including patients and healthcare workers (HCWs)”, the text currently is a bit unclear about the timeframes used for defining a “contact” between individuals over each of the different “contact” types specified throughout the text. It'd be good to have a general outline of what timeframes were used for defining a “contact” in each case, and if different timeframes are used it'd be helpful to outline why.

Reviewer #1 (Recommendations for the authors):

I believe that the current impact statement originating from this work is not completely supported by the data analysis as is. The authors should more clearly demonstrate the novelty of their work.

How sure are we, based on the provided results, that their claim is supported?

– I found it particularly difficult to understand whether all patients and HCW in the different hospital wards were sampled and sequenced when positive.

– The authors did not include any sensitivity analysis with regards to the definition of patient-patient contacts; which in my experience is not free of interpretation and therefore can have an impact on the analysis.

---

## [Author Response]

Essential Revisions:1) In their study the authors analyse data from 173 high-quality SARS-CoV-2 genomes. They identified eight patient contact clusters based on these 173 SARS-CoV-2 genomes and compare their results with an analysis taking into account patient and HCW movement data which impacts the identification of individuals belonging to the contact clusters. It is based on these differences that the authors conclude that the patient and HCW movement data improve the understanding of the role of patient-HCW interactions. I believe the authors should provide a more thorough assessment of this claim. I would like to note that their endeavour is, as such, not that novel and that researchers have looked into identifying transmission paths based on genomic surveillance with or without contextual information (see references included in the manuscript) also highlighting the importance of issues related to gathering the data needed for doing so.

We thank the reviewers for this comment, and agree that our original impact statement would benefit from some clarification. We have amended the statement to clarify the impact of our findings more explicitly. We believe that the claims in our updated impact statement have been thoroughly investigated in our manuscript, and hope that this clarification also further highlights the novelty of our work. Moreover, we have altered the title and the Abstract of the manuscript to better reflect these clarifications and to more clearly define the emphasis of the investigations in our manuscript. (Added text is underlined & removed text indicated with a strikethrough).

Manuscript Title:

“Genomic sequencing and healthcare dynamics track of nosocomial SARS-CoV-2 transmission”

Impact statement:

“We characterise the genomic divergence of SARS-CoV-2 in healthcare-associated outbreaks, demonstrating that the inclusion of healthcare workers (HCWs) into contact networks identifies additional links in SARS-CoV-2 transmission pathways.”

Abstract:

“Incorporation of HCW location further increased the number of individuals within PCCs and identified additional links in SARS-CoV-2 transmission pathways. Patients within PCCs carried viruses more genetically identical to HCWs in the same ward location and improved understanding of the role of patient-HCW interactions.”

We also acknowledge that other approaches to identify pathogen outbreaks exist, and that healthcare workers have been known to contribute to nosocomial outbreaks. We have added information to the Introduction to reflect this (see below). We also note that the rate of mutation / sequence divergence and the mechanism of transmission is specific to individual pathogens, and therefore investigations in the context of SARS-CoV-2, as described in our study, adds to the current literature and will provide informative knowledge for the accurate tracking and prevention of SARS-CoV-2 nosocomial outbreaks.

“Since the degree to which different groups (patients, HCW) propagate SARS-CoV-2 transmission remains uncertain, the utility of screening approaches to prospectively prevent nosocomial spread is difficult to evaluate. […]Here, we have integrated viral genome sequencing with patient admissions records and staff workplace information to investigate SARS-CoV-2 nosocomial outbreaks.”

2) It is not clear from the manuscript whether truly all patients had samples that were subject to genomic sequencing. How (un)likely is the exclusion of data on patients and HCW that were not sequenced and what impact might be expected from their omission?

We thank the reviewer for raising this point for clarification. We have added some additional text to the Materials and methods section to clarify this:

“RT-qPCR positive samples were selected for SARS-CoV-2 whole-genome analyses at the Manchester Centre for Genomic Medicine (MCGM; Manchester, UK). We attempted to sequence all available SARS-CoV-2 positive samples from hospital wards highlighted by infection control surveillance officers as potential outbreaks within our sample collection period due to sudden rises in positive cases.”

Of note, some of the samples that we attempted to sequence failed to fulfil the sequencing quality criteria for inclusion in other analyses. The quality criteria is defined as “≥75% of the MN908947.3 reference genome covered by ≥10 high quality reads with at least 50 aligning nucleotides”. Samples not matching this criteria included 35 samples collected from individuals on wards highlighted as potential nosocomial outbreaks. We have added some text to the manuscript to further elaborate this point.

“44 (25%) of the 173 high-quality samples were from three hospital locations which had seen sudden rises in SARS-CoV-2 positive cases; an additional 35 samples from these locations failed to meet our quality criteria for sequencing quality (Figure 1—figure supplement 1.”

Moreover we have added additional analyses to the manuscript. These data demonstrate how these restrictions of sample inclusion in subsequent analyses may impact the reported results.

“Reducing the quality threshold for sequencing datasets (≥50% coverage at ≥10x coverage) identified the MN908947.3-3228-T-G variant in an additional 5 samples collected from this hospital location (83%, *n*=6 additional samples included with modified criteria). These data highlight that samples excluded from our analyses due to sequencing quality criteria may be missing links within SARS-CoV-2 transmission pathways.”

3) The authors indicate that their work can lead to better tracking and infection control which I believe to be true but they do not provide any specific guidance on how this can be done. This final paragraph of the Discussion may be a useful place to flesh out some ideas about how this empirical data on COVID19 spread within a real-world setting may also be useful of parameterising models aimed at predicting COVID19 spread and control efforts.

In relation to this point, we have made substantial changes to the text in the Discussion section of the manuscript. We have added a new paragraph discussing these issues in depth and provided further details in the final paragraph.

“Future work may enable additional co-factors to be considered in models for network creation such as infection control measures in place on hospital wards (e.g. personal protective equipment utilized), symptomatic status of individuals, and the length, type and the proximity of physical contacts between individuals. […] Such approaches could optimise infection control management strategies, lead to targeted interventions, reduce nosocomial transmission, and ultimately prevent avoidable harm to vulnerable individuals who acquire COVID-19 whilst in the healthcare setting.”

4) Subsection “Patient Admissions and Movement”. Some additional mention of the assumptions made here (in terms of inferring contacts in this way) would be useful. Specifically, as other studies have shown that this disease is most likely to spread by short-range face-to-face contacts, it would be interesting to hear how these inferred contacts here might align to types of contact we would expect to be relevant to transmission.

We thank the reviewer for raising this important comment. We note there are a number of assumptions that are included in our approach for creating patient contact networks, including the assumption that the presence of two individuals on the same hospital ward on the same calendar day is indicative of close face-to-face contact. We note that this assumption is likely to dilute the statistical signals identified in our analyses (Figure 2) as it is likely that the inclusion of individuals that haven’t truly been in contact within contact networks would decrease the similarity in genomic variants between individuals within the same contact clusters. We acknowledge that developing systems to better characterise such contacts would increase the accuracy of the contact networks created, and potentially enable the construction of networks weighted by the types, lengths and proximity of contacts. We have added content to the manuscript to clarify and extend our discussion for all of these points.

Materials and methods:

“For each patient we identified other individuals within the cohort who were present on the same hospital wards on the same calendar day, leading to potential indirect or direct contacts between patients. […] This assumption is supported through recent meta-analyses concluding that physical distancing of less than 1 metre increases likelihood of SARS-CoV-2 transmission between individuals. (Chu et al., 2020)”

Discussion:

Please see author response to comment #3.

“While the use of digital contact tracing is difficult in hospital environments, track and tracing smartphone software could be useful to extend the characterisation of contacts between individuals and to understand the accuracy of the assumptions enforced in this study. (Firth et al., 2020; Ferretti et al., 2020 […] We note that this approach has imperfect assumptions but is likely to dilute rather than inflate the statistical significance of the investigations reported in this study (Figures 1-3).”

5) The text reads "Pairwise viral similarity was compared between clusters of the network" but there is no mention of how “clusters” within the network were defined/found. Subsection “Patient contact clusters”, again, difficult to interpret how these “clusters” were quantified from the network data. Subsection “Contact clusters including patients and healthcare workers (HCWs)”, the text currently is a bit unclear about the timeframes used for defining a “contact” between individuals over each of the different “contact” types specified throughout the text. It'd be good to have a general outline of what timeframes were used for defining a “contact” in each case, and if different timeframes are used it'd be helpful to outline why.

We again thank the reviewer for raising this point for clarification. We have added text to the Materials and methods to further expand the description of our methodologies.

“Networks of potential patient-patient and HCW contacts were constructed using the ggnet and ggplot packages in R. […] Pairwise viral similarity was compared between clusters of the network using a two-sided Wilcoxon Rank Sum test.”

Reviewer #1 (Recommendations for the authors):I believe that the current impact statement originating from this work is not completely supported by the data analysis as is. The authors should more clearly demonstrate the novelty of their work.

Please see author response to comment #1.

How sure are we, based on the provided results, that their claim is supported?– I found it particularly difficult to understand whether all patients and HCW in the different hospital wards were sampled and sequenced when positive.

Please see author response to comment #2.

– The authors did not include any sensitivity analysis with regard to the definition of patient-patient contacts; which in my experience is not free of interpretation and therefore can have an impact on the analysis.

We agree with the reviewer that this is an important point, and that the assumptions taken to infer contacts may impact accuracy of network creations and ultimately impact conclusions drawn from these analyses. We have made substantial alterations to the manuscript to better elaborate this issue. Please see author response to comment #4.